# Tax Audits, Tax Rewards and Labour Market Outcomes

Gaetano Lisi

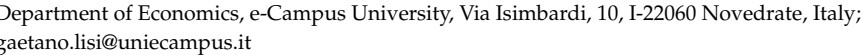

Department of Economics, e-Campus University, Via Isimbardi, 10, I-22060 Novedrate, Italy;
gaetano.lisi@uniecampus.it

**Abstract:** This theoretical paper studies the relation between tax audits and labour market outcomes (job creation and unemployment) in an economy that contemplates penalties for firms that evade taxes and rewards for firms that comply with tax rules. Intuitively, the simultaneous presence of penalty and reward amplifies the role of auditing, since tax audits allow both punishing tax-evading firms and rewarding fiscally honest firms. Indeed, the presence of tax rewards can make the effect of tax audits on firms' net profits positive. However, the effect of tax audits on labour market outcomes is ambiguous. By setting the choice of optimal fiscal policy in a different and original way, this paper is able to derive a formula for the audit rate—consistent with the budget constraint—that makes the relation between tax audits and labour market outcomes positive.

**Keywords:** tax audits; tax evasion; tax reward; job creation; unemployment

**JEL Classification:** H20; H26; H32; J64; M42

## 1. Introduction

The literature dealing with the relation between fiscal policy and tax evasion is extremely rich (Chamley 1986; Wang and Conant 1988; Slemrod 1990, 1992; Kaplow 1990; Masatoshi 1990; Schjelderup 1993; Cremer and Gahvari 1993, 1994, 1996; Jones et al. 1997; Judd 2002; Wigger 2002; Crocker and Slemrod 2005; Chen and Chu 2005; Torgler and Schaltegger 2005; Stöwhase and Traxler 2005; Kopczuk and Slemrod 2006; Hashimzade et al. 2010; Dhami and Al-Nowaihi 2010; Liu 2013; Saez 2013).

Furthermore, in the fight against tax evasion, the role of tax rewards has also been investigated (Falkinger and Walther 1991; Feld et al. 2006; Feld and Frey 2007; Kastlunger et al. 2011; Bazart and Pickhardt 2011; Murphy 2012; Brockmann et al. 2016; Fochmann and Kroll 2016). If the penalty is an economic disincentive for tax evasion, then tax reward should be an economic incentive for tax compliance (Falkinger and Walther 1991). Actually, the simultaneous presence of penalty and reward makes the tax system fairer (Lisi 2022a, 2022b).

Tax evasion and the issue of optimal fiscal policy also appear in economic growth models. Chen (2003) includes tax evasion into a standard AK growth model with public capital and studies the effect of three tax policies (cost of tax enforcement, punishment–fines and tax auditing) on both tax evasion and economic growth. It finds that the three fiscal policies are able to discourage tax evasion, but the effect on economic growth is small. Economides et al. (2020) study the properties of the optimal tax policy in a version of the neoclassical growth model where both households and firms can choose to under-report their incomes. They find that the type of tax evasion under consideration (under-report of labour income, under-report of capital income and under-report of sales) is crucial to the properties of optimal tax policy.

This theoretical paper, instead, focuses on the effect of tax evasion and fiscal policy on labour market outcomes (i.e., job creation and unemployment) in an economy that contemplates penalties for firms that evade taxes and rewards for firms that comply with tax rules. Precisely, the paper develops a modified version of the benchmark macroeconomic

model of the labour market, i.e., the search and matching model (Pissarides 2000, 2011), where tax reward is introduced in the simplest form of a "negative penalty", namely, a monetary reward for firms that comply with tax rules.

The simultaneous presence of penalty and reward amplifies the role of tax audits, since tax audits allow both punishing tax-evading firms and rewarding fiscally honest firms. Indeed, an increase in tax auditing increases firms' net profits if tax rewards are high enough. In Chen (2003), an increase in tax auditing reduces tax evasion only if the cost of tax enforcement is not too high.

However, the effect of tax audits on labour market outcomes is, a priori, ambiguous. In order to clarify that relation, the role of the tax authority is also introduced into the model. The term 'tax authority' should be understood in a broad sense, namely, the authority that decides, enforces and audits tax rules (the 'fiscal policy maker'). Thus, it also includes the key role of the government that decides the tax rules. Hence, the tax authority cares (must care) about job creation and unemployment.

By setting the tax authority's problem in a different and original way—namely, the function to be maximised is the evolution of (un)employment instead of social welfare—this paper is able to derive the level of tax audits that maximises job creation and is consistent with the budget constraint.

The rest of this theoretical paper is organised as follows. Section 2 reviews the literature on tax evasion and fiscal policy and shows the connection with the labour market. Section 3 presents the theoretical model and describes the equilibrium in the labour market. Section 4 studies the tax authority's problem and derives the optimal level of tax audits. Finally, Section 5 concludes the work and summarises the main fiscal policy implications.

## 2. Literature Review

The relationship between tax evasion/tax compliance and (optimal) fiscal policy is at once complex, challenging and fascinating (Torgler 2008).

The theory of tax evasion started with the seminal works by Allingham and Sandmo (1972), Srinivasan (1973) and Yitzhaki (1974), whereas the theory of optimal tax policy started with the seminal work by Sandmo (1981). Basically, the theory of optimal tax policy aims at giving suggestions on how to plan an efficient tax system such that the cost of taxation is reduced in the best possible way (see, e.g., Slemrod 1990).

The influential works by Allingham and Sandmo (1972), Srinivasan (1973) and Yitzhaki (1974), however, do not discuss the theory of optimal tax policy (Mirrlees 1971; Diamond and Mirrlees 1971a, 1971b; Jones et al. 1997; Chamley 1986; Judd 2002; Kopczuk and Slemrod 2006; Saez 2013).

Starting from the seminal work by Sandmo (1981), instead, several papers have introduced the social welfare objective of governments and tax administrations in models of tax evasion (see, e.g., Kaplow 1990; Masatoshi 1990; Cremer and Gahvari 1993, 1994, 1996; Schjelderup 1993; Wigger 2002; Dhami and Al-Nowaihi 2010; Liu 2013; Lisi 2015).

Of course, different types of taxes can be evaded. For example, Gordon and Nielsen (1997) allow for evasion of both value-added tax (VAT) and cash-flow income tax. Without tax evasion, taxes should have similar behavioural and distributional consequences, while the available means of evasion can be very different among taxes. In general, in planning an efficient tax system, a government should use different types of taxes, relying more on whichever tax is harder to evade. In a model calibrated to Denmark, Gordon and Nielsen (1997) find that value-added taxes are harder to evade than cash-flow income taxes.

However, when tax evasion is feasible, the normative policy implications (namely, "what should the tax authority do?"), derived from the optimal fiscal policy may be misleading (Cremer and Gahvari 1993, 1996; Boadway et al. 1994; Gueth and Sausgruber 2004; Richter and Boadway 2005). According to Gueth and Sausgruber (2004), in particular, such distortions can be avoided if the optimal taxation theory accounts for basic insights from behavioural economics.

For this reason, another important strand of literature focused on the effect of the intrinsic motivation to pay taxes (the so-called "tax morale") on tax compliance (see, e.g., Slemrod 1992; Feld and Frey 2002; Torgler 2007; Frey and Torgler 2007; Torgler and Schneider 2009; Cummings et al. 2009; Kirchler et al. 2010; Alm and Torgler 2011; Halla 2012; Molero and Pujol 2012; Castañeda 2019). The general conclusion of this stream of literature is the usefulness of going beyond a deterrence approach.

Moreover, the role of a further component of fiscal policies for increasing tax compliance, namely tax reward, has been investigated (see, e.g., Feld et al. 2006; Feld and Frey 2007; Kastlunger et al. 2011; Bazart and Pickhardt 2011; Murphy 2012; Brockmann et al. 2016; Fochmann and Kroll 2016; Lisi 2022a, 2022b).

Within this evergreen literature, the theoretical contribution of this paper is anything but trivial. Tax evasion, of course, is closely related to the phenomenon of shadow economy (an increase in the shadow income increases tax evasion). In turn, shadow economy is closely related to the phenomenon of unemployment (undeclared work often concerns unemployed workers). Hence, the use of the benchmark macroeconomic model of the labour market becomes very useful. Additionally, this work considers all the main benefits and costs of tax evasion/tax compliance, including tax rewards. Indeed, tax audits and tax rewards can contribute to building a "fiscal culture" (tax morale) in an economy where unemployment is very high and tax evasion (shadow economy) is widespread (and often tolerated). Rewarding honest taxpayers and punishing tax evaders, in fact, should be a basic principle of any democratic and modern society.

Finally, there are studies on the relationship between fiscal policy and labour market outcomes (Gomes 2010; Bova et al. 2014; Stepanyan and Leigh 2015). However, as far as we are aware, this is the first work that addresses the interplay between all fiscal policy variables (including tax rewards) and the two main indicators of labour market performance (job creation and unemployment).

## 3. The Theoretical Model

This section combines the benchmark model of the labour market (Pissarides 2000, 2011) with the core of tax evasion analysis (Sandmo 2005). Furthermore, in the spirit of Falkinger and Walther (1991), the tax evasion analysis also includes a monetary reward for tax compliance.

### 3.1. A Basic Search and Matching Model of the Labour Market

The key feature of a basic search and matching model (Pissarides 2000) is the "matching function", which expresses the number of jobs ($m$) as a function of firms' job vacancies ($v$) and unemployed workers ($u$). In this model, the matching function takes the usual form of a Cobb–Douglas function with constant returns to scale, viz.:

$$m = m(v, u) = v^{1-\alpha} \cdot u^{\alpha}$$

where $0 < \alpha < 1$ is the unemployment elasticity. From the matching function, it is straightforward to find:

- the probability of filling a job vacancy, viz.: $\left(\frac{m}{v}\right) = v^{-\alpha} \cdot u^{\alpha} \equiv \theta^{-\alpha}$, with $\frac{\partial\left(\theta^{-\alpha}\right)}{\partial\theta} < 0$;

- the probability of finding a job: $\left(\frac{m}{u}\right) = v^{1-\alpha} \cdot u^{\alpha-1} \equiv \theta^{1-\alpha}$, with $\frac{\partial\left(\theta^{1-\alpha}\right)}{\partial\theta} > 0$;

where $\theta \equiv \frac{v}{u}$ represents the so-called "labour market tightness or frictions". Intuitively, the probability of filling a job vacancy decreases in the ratio between job vacancies and unemployment; whereas, the probability of finding a job increases in the ratio between job vacancies and unemployment.

In order to find the equilibrium value of $\theta$, it needs to introduce the present value of an operative one-job firm ($r \cdot J$) and the present value of a firm' job vacancy ($r \cdot V$):

$$r \cdot J = \pi + \delta \cdot [V - J] \tag{1}$$

$$r \cdot V = -\psi + \theta^{-\alpha} \cdot [J - V] \tag{2}$$

where $r$ is the real interest rate, $\pi$ is the firm's net profit, $\psi$ is the cost of opening (and maintaining) a job vacancy and $\delta$ is the exogenous job destruction rate (the firm's dismissal rate and/or the worker's resignation rate). Equations (1) and (2) state that an operative firm ($J$) becomes a job vacancy ($V$) at the job destruction rate $\delta$, whereas a job vacancy becomes an operative firm at the matching rate $\theta^{-\alpha}$ (the probability of filling a job vacancy). By combining Equations (1) and (2) under the so-called *Job Creation condition*, namely the condition $V = 0$, it is straightforward to obtain the equilibrium value of labour market tightness ($\theta = \theta^*$):

$$\frac{\pi}{(r+\delta)} = \theta^{\alpha} \cdot \psi$$

$$\overset{yields}{\rightarrow} \theta^* = \left[ \frac{\pi}{\psi \cdot (r+\delta)} \right]^{\frac{1}{\alpha}} \tag{3}$$

Basically, the condition $V = 0$ implies that, in equilibrium, all the profit opportunities have been exploited and, thus, it is no longer convenient for a firm to open a further job vacancy.

Since $\theta^*$ derives from the job creation condition, we refer to it as "job creation". Intuitively, job creation is a positive function of the firm's net profit ($\pi$), while it is a negative function of both the cost of opening (and maintaining) a job vacancy ($\psi$) and the "total" discount rate ($r + \delta$).

### 3.2. Firm's Net Profit, Penalty, Tax Reward and Tax Audits

The net profit ($\pi$) of a firm that can evade taxes or comply with tax rules is the following:

$$\pi = y - \tau \cdot y^D - \rho \cdot \left( \gamma^{-1} \cdot \tau \cdot s \right) + \rho \cdot (b \cdot \gamma) - \varphi \tag{4}$$

where $y$ is the exogenous (true) net income (revenues net of wages); $y^D$ is the declared income (the firm's tax base); $\tau$ is the tax rate (a "linear income tax", for the sake of simplicity); $\rho$ is the audit rate; $s \equiv \left( y - y^D \right) \geq 0$ is the evaded income; $b > 0$ is the monetary tax reward; $\gamma \equiv \frac{y^D}{y}$ is the reward rate; $\gamma^{-1} \equiv \frac{y}{y^D}$ is the multiplier of taxation (the weight of penalty); and $\varphi$ the concealment cost of the evaded income.[1]

Note that $s \equiv \left( y - y^D \right)$ is the shadow economy (income), while $\tau \cdot s$ is tax evasion.

Regarding $\gamma^{-1}$ and $\gamma$, a fair penalty should be assessed on the level of tax evasion (and it should be always higher than taxation); additionally, a right reward should be assessed on the level of declared income. Hence, the higher the declared income, the higher the $\gamma$ and the lower the $\gamma^{-1}$, and vice versa. At the limit, when $y = y^D$ (and thus $s = 0$), the full monetary reward is received; on the other hand, when $y^D \rightarrow 0$, the penalty becomes very high, thus cancelling out profits entirely, i.e., $\lim\limits_{y^D \to 0} \pi \to 0$.

The firm chooses the level of declared income, $y^D = \left( y^D \right)^*$, that maximises the net profit (for mathematical details, see Appendix A):

$$\left( y^D \right)^* = \frac{y}{\left( \frac{1}{\rho} - \frac{b}{y \cdot \tau} \right)^{\frac{1}{2}}} \tag{5}$$

where $\left( y^D \right)^*$ is positive for $y > \frac{b \cdot \rho}{\tau}$. From Equation (5), the shadow income ($s$) is also obtained. Intuitively, $\frac{\partial \left( y^D \right)^*}{\partial b} > 0$, $\frac{\partial \left( y^D \right)^*}{\partial \tau} < 0$ and $\frac{\partial \left( y^D \right)^*}{\partial \rho} > 0$.

Furthermore, note that the effect of tax audits on firms' net profits can be positive, viz.:

$$\frac{\partial \pi}{\partial \rho} = -\left( \gamma^{-1} \cdot \tau \cdot s \right) + (b \cdot \gamma) > 0$$

if $b > \left[ \left( \gamma^{-1} \right)^2 \cdot \tau \cdot s \right]$. As a result,

**Proposition 1.** *Tax reward increases the declared income by firms and amplifies the role of tax audits. Precisely, a high tax reward can make the effect of tax audits on firms' net profits positive.*

Looking at Equation (3), therefore, the presence of taw rewards could trigger a virtuous circle, i.e., the higher the tax reward, the higher the declared income by firms, the higher the firms' net profits and the higher job creation. In turn, an increase in the declared income by firms increases tax rewards and, thus, the virtuous circle starts again.

*3.3. Labour Market Outcomes and Fiscal Policies*

By normalising the labour force to the unit, the evolution of unemployment over time ($t$) is given by:

$$\dot{u} \equiv \frac{du}{dt} = [\delta \cdot (1 - u)] - \left[ \theta^{1-\alpha} \cdot u \right] \tag{6}$$

where $[\delta \cdot (1 - u)]$ is the inflows into unemployment, i.e., the employed workers $(1 - u)$ who lose their jobs at the job destruction rate $(\delta)$, whereas $\left[ \theta^{1-\alpha} \cdot u \right]$ is the unemployment outflows, namely, the unemployed workers $(u)$ who find a job at the probability of finding a job $(\theta^{1-\alpha})$.

Thus, in the steady state (with $\dot{u} = 0$), the equilibrium unemployment $(u^*)$ is given by:

$$u^* = \frac{\delta}{\delta + \theta^{1-\alpha}} \tag{7}$$

From Equation (7), the negative relation between unemployment and job creation clearly emerges, since the probability of finding a job $(\theta^{1-\alpha})$ positively depends on job creation $(\theta)$.

However, the effect of tax audits on job creation and unemployment is, a priori, ambiguous (for mathematical details, see Appendix B). Precisely,

**Proposition 2.** *The effect of tax audits on job creation and unemployment depends on penalty and reward that, in turn, depend on the budget constraint, viz.:*

$$\underbrace{\left( \tau \cdot y^D \right) + \rho \cdot \left( \gamma^{-1} \cdot \tau \cdot s \right)}_{revenue} - \underbrace{c(\rho) - \rho \cdot b \cdot \gamma - g}_{expenditure} \geq 0 \tag{8}$$

*where g is the public spending per capita (the needs of the public sector) and $c(\rho)$, with $\frac{dc(\rho)}{d\rho} > 0$, is the auditing cost, namely, the administrative cost to the tax authority to increase tax audits.*

Actually, an increase in tax audit rate $(\rho)$ increases both the penalty component $\left( \gamma^{-1} \cdot \tau \cdot s \right)$, which increases state revenues, and the reward component $(b \cdot \gamma)$, which increases public spending. As a result, its net effect is, a priori, ambiguous.

## 4. Fiscal Policy

In order to define the rate of tax audits that is harmless for labour market outcomes and consistent with the budget constraint, it is needed to introduce the tax authority's problem.

Precisely, this section assumes that the (benevolent and forward-looking) tax authority maximises labour market performance, i.e., the evolution of employment:

$$\epsilon = 1 - u$$

under the (balanced) budget constraint:[2]

$$\begin{cases} max \left[ \int_0^\infty \left[ \theta^{1-\alpha} \cdot (1 - \epsilon) - \delta \cdot \epsilon \right] \cdot e^{-r \cdot t} dt \right] \\ s.t. \ g = \left( \tau \cdot y^D \right) + \rho \cdot \left( \gamma^{-1} \cdot \tau \cdot s \right) - c(\rho) - \rho \cdot b \cdot \gamma \end{cases} \tag{9}$$

the fiscal policy variables ($\tau$, $\rho$ and $b$) are the 'control variables' while the public spending ($g$) is the 'state variable'.

The tax authority's problem (9) is specified in a different and somehow original way with respect to the standard social planner problem, where some version of social welfare is maximised.[3] As analysed in Section 3.3., fiscal policy variables directly affect job creation and, thus, unemployment. Furthermore, an increase in unemployment could rise shadow income and tax evasion, since unemployed workers can find a job in the shadow economy. Hence, it makes sense for the tax authority or the government (the 'fiscal policy maker') to aim for the maximisation of employment (the minimisation of unemployment).

The solution to the dynamic optimisation problem gives the rate of tax audits ($\overline{\rho}$) that makes $\frac{\partial(\theta^{1-\alpha})}{\partial\rho} = 0$ (for mathematical details, see Appendix C):

$$\overline{\rho} = \left[\frac{(\gamma^{-1}\cdot\tau\cdot s) - b\cdot\gamma}{\omega}\right]^{\frac{1}{\omega-1}} \tag{10}$$

where $\omega > 0$ is the elasticity of the auditing cost function (see again Appendix C).

Consequently, the "optimal" level of tax audits ($\rho^*$) should be always higher than $\rho^*$:

$$\rho^* > \overline{\rho}$$

since the "optimal" level of tax audits ($\rho^*$) is the level that makes the relation between tax audits and labour market outcomes positive, i.e., $\frac{\partial\theta^*}{\partial\rho} > 0$ and, thus, $\frac{\partial u^*}{\partial\rho} < 0$.

Note that this also implies an increase in official employment and a decrease in shadow employment, since $\frac{\partial(y^D)^*}{\partial\rho} > 0$ in Equation (5) and, thus, $\frac{\partial s^*}{\partial\rho} < 0$. In short, a high tax audit decreases both unemployment and shadow income. It follows that official employment should increase more than the decrease in shadow employment. Employment, therefore, will migrate from "the grey zone into daylight". Unfortunately, job migration is a very 'complex' socioeconomic dynamic that this simple model is not able to directly catch.

Therefore, once the tax authority detects tax compliance ($y^D$), the rates/weights $\gamma^{-1}$ and $\gamma$ are obtained. Given $s$, $\gamma^{-1}$ and $\gamma$, the ratio between taxation ($\tau$) and tax reward ($b$) consistent with Equation (10) is given by:

$$\frac{\tau}{b} \geq \frac{\gamma}{\gamma^{-1}\cdot s}$$

Finally, the tax authority should fix $\rho^* > \overline{\rho}$. Accordingly,

**Proposition 3.** *Once a benchmark level of auditing is identified, optimal tax audits should be fixed at a higher level rather than at a lower level.*

A high tax audit rate, indeed, could be the best fiscal policy in "bad" scenarios. First of all, when tax evasion is a widespread phenomenon, penalty, reward and tax audits can contribute to build a "fiscal culture", since rewarding honesty (in a broad sense) and punishing corrupt and criminal behaviour should be a basic principle of any democratic and modern society. Furthermore, in an "abnormal" situation (where there are too many tax evaders), tax audits should be high, and "normality" (complying with the tax rules) should be rewarded.

## 5. Conclusions

The literature on the optimal fiscal policy in the presence of tax evasion is rich and important. However, as far as we are aware, there are no studies that address, at the same time, the interplay between fiscal policy, tax evasion and labour market outcomes (job creation and unemployment) in the presence of tax rewards.

Therefore, this theoretical paper introduces into the benchmark model of the labour market, i.e., the search and matching model, the core of tax evasion analysis and the possibility of a monetary reward for firms that comply with tax rules.

The presence of tax rewards amplifies the role of auditing. Precisely, tax audits increase the declared income by firms and, when tax rewards are high, tax auditing also increases firms' net profits.

Meanwhile, at the macroeconomic level, the effect of tax audits on job creation and unemployment is, a priori, ambiguous and depends on both penalty and reward.

In order to clarify that relation, the choice of optimal fiscal policy is fixed in a different and original way. Precisely, the tax authority (the 'fiscal policy maker') maximises labour market performance, i.e., the evolution of employment, under the balanced budget constraint. Unemployment, indeed, is closely related to shadow economy, which, in turn, is closely related to tax evasion.

Eventually, the model is able to derive a formula for the audit rate—consistent with the budget constraint—that makes the relation between tax audits and labour market outcomes positive. From an economic policy point of view, once a benchmark level of auditing is identified, the tax authority should fix the audit rate at a higher level rather than at a lower level.

Concisely, there could be a positive relation between tax audits, tax compliance and labour market outcomes. In countries (like Italy)—where unemployment is very high, shadow economy is widespread and often tolerated—tax audits and tax rewards can contribute to building a "fiscal culture", since rewarding honest taxpayers and punishing tax evaders should be a basic principle of any democratic and modern society. Unfortunately, however, tax rewards are poorly considered, and tax audits are either too low or ineffective. This theoretical paper, indeed, shows that high tax audits, in the presence of tax rewards, can reduce both tax evasion and unemployment. Of course, in the short run, this policy increases public expenditure, but in the long run, the increase in declared income will increase state revenues and the increase in the net profit of firms will increase job creation. Without forgetting that, when a "fiscal culture" is realised, a virtuous circle can arise.

**Funding:** This research received no external funding.

**Institutional Review Board Statement:** Not applicable.

**Informed Consent Statement:** Not applicable.

**Data Availability Statement:** Data are not available (theoretical research).

**Acknowledgments:** The author is indebted to the two anonymous referees for the many and helpful comments and suggestions.

**Conflicts of Interest:** The authors declare no conflict of interest.

## Appendix A

The solution to the maximisation problem of a firm requires the usual first-order-condition:

$$\frac{d\pi}{dy^D} = \frac{d\left\{ y - \tau \cdot y^D - \rho \cdot \left[ \frac{y \cdot \tau \cdot (y - y^D)}{y^D} \right] + \rho \cdot \left( b \cdot \frac{y^D}{y} \right) - \varphi \right\}}{dy^D} = 0$$

thus obtaining, after some algebraic steps, Equation (5):

$$-\tau - \rho \cdot \tau \cdot \left( -\frac{y}{y^D} \right)^2 + \frac{\rho \cdot b}{y} = 0$$

$$\rho \cdot \tau \cdot \left( \frac{y}{y^D} \right)^2 + \frac{\rho \cdot b}{y} = \tau$$

$$\left(\frac{y}{y^D}\right)^2 = \frac{\left(\tau - \rho \cdot \frac{b}{y}\right)}{\rho \cdot \tau}$$

$$\left(\frac{y}{y^D}\right)^2 = \frac{1}{\rho} - \frac{b}{y \cdot \tau}$$

$$\frac{y}{y^D} = \left(\frac{1}{\rho} - \frac{b}{y \cdot \tau}\right)^{\frac{1}{2}}$$

$$\left(y^D\right)^* = \frac{y}{\left(\frac{1}{\rho} - \frac{b}{y \cdot \tau}\right)^{\frac{1}{2}}}$$

## Appendix B

The labour market equilibrium is characterised by the job creation Equation (3) and the unemployment Equation (7):

$$\theta^* = \left[\frac{\pi}{\psi \cdot (r + \delta)}\right]^{\frac{1}{\alpha}}$$

$$u^* = \frac{\delta}{\delta + \theta^{1-\alpha}}$$

with

$$\frac{\partial u^*}{\partial \theta^*} = (-1) \cdot \frac{\delta}{[\delta + \theta^{1-\alpha}]^2} \cdot (1 - \alpha) \cdot \theta^{-\alpha} < 0$$

since $0 < \alpha < 1$. It follows that:

$$\frac{\partial \theta^*}{\partial \tau} = \frac{1}{\alpha} \cdot \left(\frac{\pi}{\psi \cdot (r + \delta)}\right)^{\frac{1}{\alpha} - 1} \cdot \left(-y^D - \gamma^{-1} \cdot s\right) < 0$$

thus,

$$\frac{\partial u^*}{\partial \tau} > 0$$

and,

$$\frac{\partial \theta^*}{\partial b} = \frac{1}{\alpha} \cdot \left(\frac{\pi}{\psi \cdot (r + \delta)}\right)^{\frac{1}{\alpha} - 1} \cdot \rho \cdot \gamma > 0$$

thus,

$$\frac{\partial u^*}{\partial b} < 0$$

Meanwhile, the effect of tax audits on job creation and unemployment is, a priori, ambiguous:

$$\frac{\partial \theta^*}{\partial \rho} = \frac{1}{\alpha} \cdot \left(\frac{\pi}{\psi \cdot (r + \delta)}\right)^{\frac{1}{\alpha} - 1} \cdot \left(-\gamma^{-1} \cdot \tau \cdot s + b \cdot \gamma\right)$$

Precisely, it depends on the sign of $\left(-\gamma^{-1} \cdot \tau \cdot s + b \cdot \gamma\right)$.

## Appendix C

Let $\lambda$ be the costate variable, i.e., the shadow value of a marginal decrease in the public spending at time $t$, so that the Hamiltonian ($H$) is:

$$H = \left\{\left[\theta^{1-\alpha} \cdot (1 - \epsilon) - \delta \cdot \epsilon\right] + \lambda \cdot \left[\left(\tau \cdot y^D\right) + \rho \cdot \left(\gamma^{-1} \cdot \tau \cdot s\right) - c(\rho) - \rho \cdot b \cdot \gamma\right]\right\} \cdot e^{-r \cdot t}$$

The solution to this dynamic optimisation problem requires that:[4]

$$\frac{\partial H}{\partial \tau} = 0 \overset{yields}{\to} \frac{\partial(\theta^{1-\alpha})}{\partial \tau} \cdot (1 - \epsilon) + \lambda \cdot \left[ y^D + \rho \cdot \left( \gamma^{-1} \cdot s \right) \right] = 0$$

$$\overset{yields}{\to} \left( -\frac{\partial(\theta^{1-\alpha})}{\partial \tau} \right) \cdot (1 - \epsilon) = \lambda \cdot \left[ y^D + \rho \cdot \left( \gamma^{-1} \cdot s \right) \right] \qquad \text{(A1)}$$

$$\frac{\partial H}{\partial b} = 0 \overset{yields}{\to} \frac{\partial \partial(\theta^{1-\alpha})}{\partial b} \cdot (1 - \epsilon) - \lambda \cdot \rho \cdot \gamma = 0$$

$$\overset{yields}{\to} \frac{\partial(\theta^{1-\alpha})}{\partial b} \cdot (1 - \epsilon) = \lambda \cdot \rho \cdot \gamma \qquad \text{(A2)}$$

$$\frac{\partial H}{\partial \rho} = 0 \overset{yields}{\to} \frac{\partial(\theta^{1-\alpha})}{\partial \rho} \cdot (1 - \epsilon) + \lambda \cdot \left[ \left( \gamma^{-1} \cdot \tau \cdot s \right) - \frac{dc(\rho)}{d\rho} - b \cdot \gamma \right] = 0$$

$$\overset{yields}{\to} \frac{\partial(\theta^{1-\alpha})}{\partial \rho} \cdot (1 - \epsilon) = \lambda \cdot \left[ \frac{dc(\rho)}{d\rho} + b \cdot \gamma - \left( \gamma^{-1} \cdot \tau \cdot s \right) \right] \qquad \text{(A3)}$$

By using the optimality conditions (A3) and (A2), we obtain:

$$\frac{\partial(\theta^{1-\alpha})}{\partial \rho} = \frac{\left[ \frac{dc(\rho)}{d\rho} + b \cdot \gamma - \left( \gamma^{-1} \cdot \tau \cdot s \right) \right]}{\rho \cdot \gamma} \cdot \frac{\partial(\theta^{1-\alpha})}{\partial b}$$

Note that now the sign of $\frac{\partial(\theta^{1-\alpha})}{\partial \rho}$ also depends on the marginal effect of the auditing cost, viz.:

$$\frac{dc(\rho)}{d\rho} + b \cdot \gamma - \left( \gamma^{-1} \cdot \tau \cdot s \right)$$

This finding is robust, since it does not change when the optimality conditions (A3) and (A1) are used:[5]

$$\frac{\partial(\theta^{1-\alpha})}{\partial \rho} = \frac{\left[ \frac{dc(\rho)}{d\rho} + b \cdot \gamma - \left( \gamma^{-1} \cdot \tau \cdot s \right) \right]}{[y^D + \rho \cdot (\gamma^{-1} \cdot s)]} \cdot \left( -\frac{\partial(\theta^{1-\alpha})}{\partial \tau} \right)$$

with $\left( -\frac{\partial(\theta^{1-\alpha})}{\partial \tau} \right) > 0$, since $\frac{\partial(\theta^{1-\alpha})}{\partial \tau} < 0$.

By assuming that $c(\rho) = \rho^\omega$, with a positive elasticity $\omega > 0$, we obtain the value of tax audits that makes $\frac{dc(\rho)}{d\rho} + b \cdot \gamma - \left( \gamma^{-1} \cdot \tau \cdot s \right) = 0$ and, thus, $\frac{\partial(\theta^{1-\alpha})}{\partial \rho} = 0$, viz.:

$$\omega \cdot \rho^{\omega-1} + b \cdot \gamma - \left( \gamma^{-1} \cdot \tau \cdot s \right) = 0$$

$$\overset{yields}{\to} \bar{\rho} = \left[ \frac{\left( \gamma^{-1} \cdot \tau \cdot s \right) - b \cdot \gamma}{\omega} \right]^{\frac{1}{\omega-1}}$$

with $\lim\limits_{y^D \to 0} \bar{\rho} \to \infty$ and $\lim\limits_{y^D \to y} \bar{\rho} \to 0$.

Recall that $\frac{\partial(\theta^{1-\alpha})}{\partial \theta} > 0$; thus, the sign of $\frac{\partial \theta}{\partial \rho}$ is equal to the sign of $\frac{\partial(\theta^{1-\alpha})}{\partial \rho}$.

## Notes

[1] As in Kolm and Larsen (2019), the concealment cost of tax evasion is just a parameter. Chen (2003) and Economides et al. (2020) assume that this cost depends on the level of income and the degree of tax evasion. However, that assumption would only complicate the mathematics, but it would not change the key result of the following Proposition 1.

[2] For the sake of clarity, we neglect the time reference of the variables. Of course, with respect to Equation (6), the inflows into unemployment become the employment outflows and the unemployment outflows become the inflows into employment.

[3] For example, in the standard search and matching model (Pissarides 2000), the tax authority maximises the social welfare function (which, for an infinitely lived economy, is equal to the total net profits minus vacancy costs and plus the benefit of being unemployed) under the constraint represented by the evolution of unemployment.

[4] There is also the optimality condition with respect to the state variable, namely $-\frac{\partial H}{\partial g} = \frac{d[\lambda(t) \cdot e^{-r \cdot t}]}{dt} = \dot{\lambda} - \lambda \cdot r$.

[5] We need to always use condition (C3) because the ambiguous relation concerns the sign of $\frac{\partial(\theta^{1-\alpha})}{\partial \rho}$.

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
