# Peer review of "Tax Audits, Tax Rewards and Labour Market Outcomes"

_economies, doi:10.3390/economies11020060_

Round 1
Reviewer 1 Report
Referee report for “Tax audits and labour market outcomes”
This paper examines the effects of tax audits on labour market outcomes (job creation and unemployment), when there are penalties for the tax evaded firms and rewards for firms that comply, and finds that these effects are ambiguous, depending on fiscal policy.
In my opinion this is a very interesting and well-established paper. However, I have some minor comments.
1) The introduction is rather “poor”. I missed a brief discussion about the main pillars that connect fiscal policy and tax evasion and how this paper is differentiated compared to the relevant literature. Thus, the authors should elaborate more on this issue, either including one or two paragraphs with a short literature review (not just mentioning the papers as in the first paragraph) and adding a separate section with this discussion. In any case, they should state clearly how their paper differs from the literature.
2) Moreover, the authors should include some additional papers that examine the relationship between optimal policy and tax evasion. For instance, Economides et al., 2020, and Gordon and Nielsen (1997). The latter allows for evasion of both VAT and income taxes in a model calibrated to Denmark, while the first one examines how the optimal tax policy mix changes under the presence of tax evasion. I believe that this discussion would be useful to be included as an additional paragraph in the introduction section, discussion briefly the relationship between optimal policy and tax evasion.
3) A well-established paper that models tax evasion is that of Chen (2003). I suggest that the authors could explain briefly how the introduction of tax evasion in this paper differs from the one used in Chen (2003).
For example, the authors suggest that the concealment cost of tax evasion is just a parameter. However Chen (2003) and Economides et al (2020) assume that this cost depends on the level of income and the degree of tax evasion.
4) I think that the paper is rather weak in explaining its (interesting) results. For instance, the authors find that the effect of tax audits on job creation and unemployment is ambiguous. What are the channels behind this result? In general, I suggest that the authors should elaborate a bit more on the intuition of their key results.
References
Chen, B. L. (2003). Tax evasion in a model of endogenous growth. Review of Economic Dynamics, 6, 381–403.
Economides, G., Philippopoulos, A., and Rizos, A. (2020). Optimal tax policy under tax evasion, International Tax and Public Finance, 17, 339-362.
Gordon, R., & Nielsen, S. B. (1997). Tax evasion in an open economy: Value-added vs. income taxation. Journal of Public Economics, 66(2), 173–197.
Reviewer 2 Report
The topic is certainly interesting. I personally have a problem with the whole premise of the paper.
Three are indeed studies on the relationship between tax audits and profits. Yet, there is a good reason why the relationship between tax audits and labor market outcomes is less investigated.
The economy is a complex system with multiple feedback loops. Its dynamic can hardly be captured by closed-end formulas. If the approach were relying on agent-based models, the conclusions would have been more legitimate. Labor market outcomes and tax audits are too many steps removed from each other. There are too many layers of complexity and non-linear dynamics in between ...the whole idea is barely credible….unless we take the paper as an exercise in escapist math with very little bearing on the real world.
I’m not sure what are the real-life implications, although I understand the conclusion on an intuitive level. But there is no need for all that math, and it seems to me the premise is set up so as to beg the question. It seems the economic backdrop is just a pretext for showing off the math.
Minor editing comment:
The sentence ”From a macroeconomic point of view, instead, the effect of tax audits on labour market outcomes (job creation and unemployment) is, a priori, ambiguous. In order to clarify the relation between tax audits, job creation and unemployment, the role of the tax authority is also introduced into the model.” is correct from a grammatical point of view yet it reads awkward and is contrived. Please re-phrase this...and several others like it.
Round 2
Reviewer 2 Report
The revised version brings several improvements and adjustments. While I will not comment on all of them, there is one that I find particularly interesting. In the end, the author(s) manage to give a coherent and insightful explanation for why we should expect a positive relationship between tax audits and labor market outcomes.
Yet again, I feel the main problem that I signaled earlier still remains. The assumptions and the entire set up begs the question. Everything is just-so.
The math doesn't account for the complexity of the economy and the many feedback mechanisms out there.
I can bypass all that math by appealing to one's intuition: to the extent to which tax audits are associated with a decrease in tax evasion and the grey economy, one should naturally expect an increase in official employment. Labor will migrate from the grey zone into daylight.
Minor editing comments
Make sure you correct sentences such as: "Finally, there are works dealt with the relationship between fiscal policy and labour market outcomes." line 125 in the revised manuscript...
